# Inhibition of noradrenergic signalling in rodent orbitofrontal cortex impairs the updating of goal-directed actions

Juan Carlos Cerpa[1†§], Alessandro Piccin[1†], Margot Dehove[1], Marina Lavigne[2], Eric J Kremer[2], Mathieu Wolff[1], Shauna L Parkes[1*§], Etienne Coutureau[1*§]

[1]CNRS, University of Bordeaux, Bordeaux, France; [2]Institut de Génétique Moléculaire de Montpellier, CNRS, University of Montpellier, Montpellier, France

**Abstract** In a constantly changing environment, organisms must track the current relationship between actions and their specific consequences and use this information to guide decision-making. Such goal-directed behaviour relies on circuits involving cortical and subcortical structures. Notably, a functional heterogeneity exists within the medial prefrontal, insular, and orbitofrontal cortices (OFC) in rodents. The role of the latter in goal-directed behaviour has been debated, but recent data indicate that the ventral and lateral subregions of the OFC are needed to integrate changes in the relationships between actions and their outcomes. Neuromodulatory agents are also crucial components of prefrontal functions and behavioural flexibility might depend upon the noradrenergic modulation of the prefrontal cortex. Therefore, we assessed whether noradrenergic innervation of the OFC plays a role in updating action-outcome relationships in male rats. We used an identity-based reversal task and found that depletion or chemogenetic silencing of noradrenergic inputs within the OFC rendered rats unable to associate new outcomes with previously acquired actions. Silencing of noradrenergic inputs in the prelimbic cortex or depletion of dopaminergic inputs in the OFC did not reproduce this deficit. Together, our results suggest that noradrenergic projections to the OFC are required to update goal-directed actions.

**\*For correspondence:**
shauna.parkes@u-bordeaux.fr (SLP);
etienne.coutureau@u-bordeaux.fr (EC)

[†]These authors contributed equally to this work

**Present address:** [§]Department of Experimental Psychology, University of Oxford, Oxford, United Kingdom

[§]Senior author

## Editor's evaluation

The capacity to flexibly modify our actions in order to seek goals relies upon specific brain regions and neurochemicals. Here, Cerpa et al., identify norepinephrine (but not dopamine) within the ventrolateral orbitofrontal cortex (OFC) as key to updating identity-specific action-outcome associations when environmental conditions change. These conclusions are well supported by the data and will be of interest to behavioural neuroscientists studying the function of OFC or noradrenaline signalling, as well as researchers studying associative learning more broadly.

## Introduction

Animals use their knowledge of an environment to engage in behaviours that meet their basic needs and desires. In a dynamic environment, an animal must also be able to update its understanding of the setting, particularly when the outcomes or consequences of its actions change. Numerous studies indicate that goal-directed behaviours are supported by the prefrontal cortex (PFC), and current research suggests a parcellation of functions within prefrontal regions in rodents (for reviews, see *O'Doherty et al., 2017*, *Coutureau and Parkes, 2018*). Specifically, the prelimbic region or Area 32 (A32) in *Paxinos and Watson, 2014*, of the medial PFC is needed to initially acquire goal-directed actions and learn the relationship between distinct actions and their outcomes (*Hart et al., 2018*;

*Killcross and Coutureau, 2003*; *Tran-Tu-Yen et al., 2009*), whereas the gustatory region of the insular cortex is required to recall the current value of these outcomes to guide choice between competing actions (*Balleine and Dickinson, 2000*; *Parkes and Balleine, 2013*; *Parkes et al., 2015*). In addition, the ventral (VO) and lateral (LO) subregions of the orbitofrontal cortex (OFC) are required for goal-directed actions (*Gremel and Costa, 2013*; *Gremel et al., 2016*; *Zimmermann et al., 2017*; *Rhodes and Murray, 2013*; *Fiuzat et al., 2017*; *Malvaez et al., 2019*; *Sias et al., 2021*; *Wassum, 2022*), specifically to update previously established instrumental associations (*Parkes et al., 2018*).

Behavioural flexibility also requires activity in noradrenergic (NA) neurons of the locus coeruleus (LC), which are thought to track uncertainty in the current situation (*Bouret and Sara, 2004*; *Cope et al., 2019*; *Jahn et al., 2018*; *McGaughy et al., 2008*; *Tait et al., 2007*; *Tervo et al., 2014*; *Uematsu et al., 2017*). Most notably, compelling theoretical models hypothesize that the LC interacts with the PFC to support behavioural flexibility (*Sara and Bouret, 2012*). Taken together, these data raise the intriguing possibility that LC NA innervation of the OFC might be needed to update previously established goal-directed actions (*Chandler et al., 2013*; *Agster et al., 2013*; *Sadacca et al., 2017*, *Cerpa et al., 2019*; *Cerpa et al., 2021*). To investigate this possibility, we examined whether LC NA inputs to the OFC are required to learn that a specific outcome associated with a given action has changed, and to recall this information to guide choice.

Specifically, after learning initial action-outcome (A-O) associations, rats were required to flexibly encode and use new associations during an instrumental reversal task. First, we depleted NA fibres using anti-DβH saporin (SAP) and observed a profound deficit in the ability to use the reversed A-O associations to guide choice. We then found that this deficit is not present when we depleted dopaminergic (DA) signalling in the OFC. Finally, we investigated the temporal and anatomical specificity of this effect using viral vector-mediated expression of inhibitory DREADDs. We found that silencing LC:$^{vlOFC}$, but not LC:$^{A32}$, projections impaired the ability to acquire and express the reversed instrumental contingencies. Collectively, these data suggest that LC NA projections to the OFC are likely required for both encoding and recalling the identity of an expected instrumental outcome, specifically when that identity has changed.

## Results
### Initial goal-directed learning does not require NA signalling in the OFC

We first assessed if the initial acquisition and expression of goal-directed actions requires NA signalling in the OFC using the behavioural design shown in *Figure 1A*. To deplete NA projections, rats were given bilateral injections of the toxin anti-DβH SAP targeting the ventral and lateral regions of the OFC (vlOFC). Animals from the control (CTL) group were injected with inactive IgG SAP. Rats in the Pre group were injected with either anti-DβH SAP (group Pre-SAP n=15) or inactive IgG SAP (group Pre-CTL n=14) before the initial instrumental training, during which responding on one action (A1) earned O1 (sucrose or grain pellets, counterbalanced), and responding on the other (A2) earned O2 (grain or sucrose pellets, counterbalanced). Rats in group Post were similarly trained, but were injected with either anti-DβH SAP (group Post-SAP n=15) or inactive IgG toxin (group Post-CTL n=13) following this initial stage.

We first quantified the loss of NA fibres using DβH immunostaining in the vlOFC (VO and LO) and Area 32 dorsal (A32d) and ventral (A32v), as shown in *Figure 1B–C* (for additional photomicrographs, see *Figure 1—figure supplement 1*). SAP infusion resulted in extensive NA fibres loss in the VO and LO, as revealed by a main effect of group ($F_{(1,53)}$ = 277.33, p<0.001) and region ($F_{(1,53)}$ = 12.25, p<0.01). A significant group (CTL vs. SAP) × region (VO vs. LO) interaction effect was also detected ($F_{(1,53)}$=12.41, p<0.01) and simple effect analyses revealed that NA fibres volume was greater in the VO compared to LO for the CTL group ($F_{(1,53)}$ = 23.38, p<0.001), a result consistent with previous findings (*Cerpa et al., 2019*), but this was not the case for the SAP group ($F_{(1,53)}$ = 0.00, p>0.975). Significant fibres loss was also observed in A32 ($F_{(1,53)}$ = 163.64, p<0.001; *Figure 1C*), as well as medial orbitofrontal cortex (MO; $F_{(1,49)}$ = 171.92, p<0.001), secondary motor cortex (M2; $F_{(1,53)}$ = 175.49, p<0.001), and insula cortex ($F_{(1,46)}$ = 63.5, p<0.001) (*Figure 1—figure supplement 2*).

As shown in *Figure 1D*, all rats acquired the lever pressing response, with their rate of lever pressing increasing across days ($F_{(1,53)}$ = 508.30, p<0.001). No differences were found between Pre and Post groups ($F_{(1,53)}$ = 0.96, p=0.33) or between CTL and SAP groups ($F_{(1,53)}$ = 0.287, p=0.59) and there were

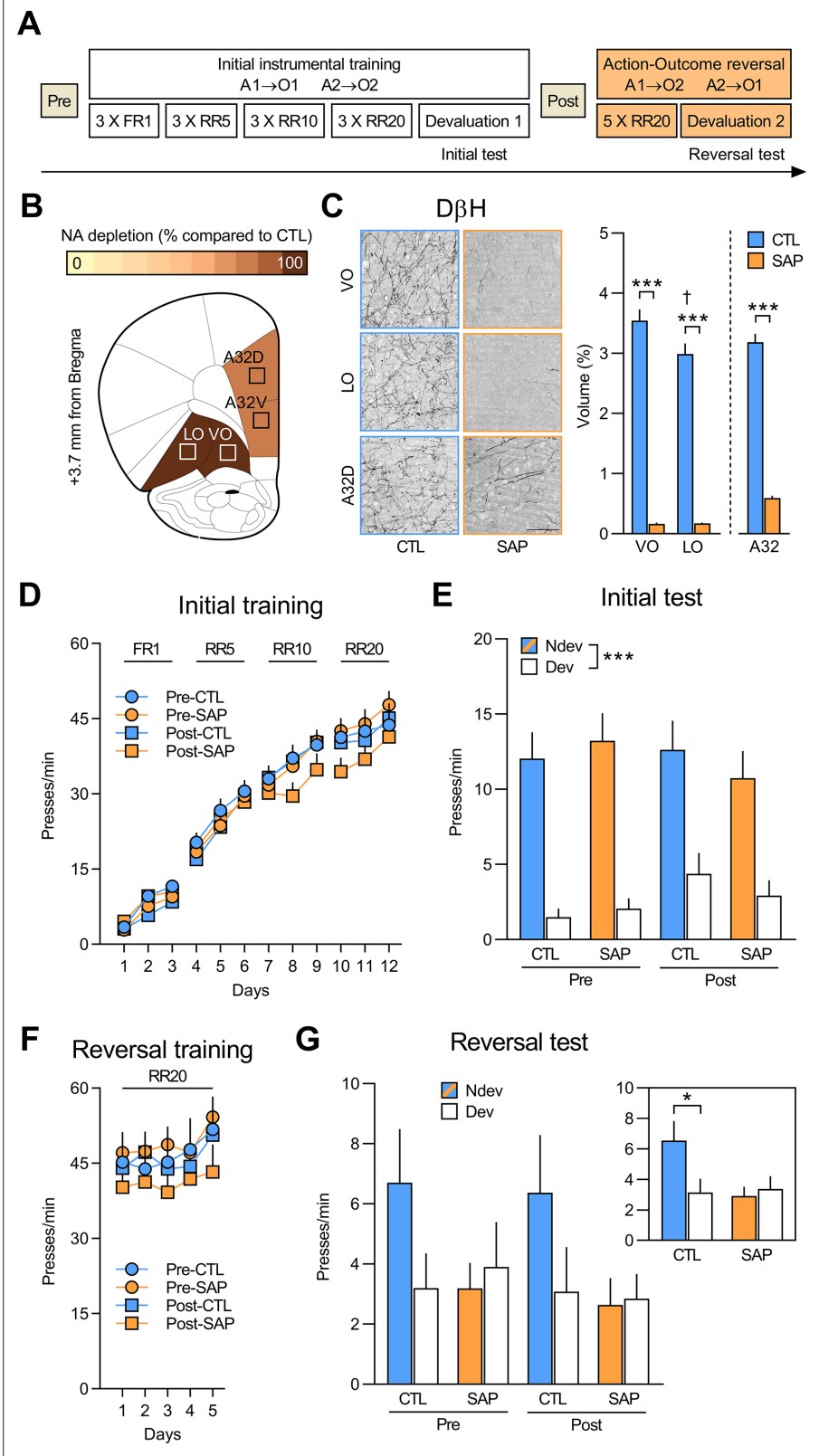

**Figure 1.** Initial goal-directed learning does not require NA signalling in the OFC. (**A**) Experimental timeline for rats injected with anti-DβH saporin (SAP) or the inactive control (CTL) toxin before (Pre) and after (Post) initial instrumental training and outcome devaluation (Pre-CTL n=14, Pre-SAP n=15, Post-CTL n=13, Post-SAP n=15). (**B**) Regions where dopamine beta hydroxylase (DβH)-positive fibres were quantified (Area 32 dorsal: A32d; Area 32

*Figure 1 continued on next page*

*Figure 1 continued*

ventral: A32v; ventral orbitofrontal cortex: VO; lateral orbitofrontal cortex: LO); schematics adapted from Figure 9 of The Rat Brain in Stereotaxic Coordinates (*Paxinos and Watson, 2014*). (**C**) Representative photomicrographs of noradrenergic (NA) depletion and DβH fibres volume (%) in VO (+3.7 mm from Bregma), LO (+3.0 mm from Bregma), and A32d (+4.4 mm from Bregma) following toxin injection. (**D**) Rate of lever pressing across initial training (A1–O1; A2–O2), collapsed across the two actions. (**E**) Initial instrumental test in extinction following satiety-induced devaluation (Ndev: non-devalued; Dev: devalued). (**F**) Rate of lever pressing across reversal training (A1–O2; A2–O1), collapsed across the two actions. (**G**) Reversal instrumental test in extinction following satiety-induced devaluation. The inlet shows data grouped for CTL (Pre and Post) and SAP groups (Pre and Post). Data are presented as mean + SEM. *p<0.05, ***p<0.001, †p<0.05 LO vs. VO CTL group. Scale bars: 100 µm. A1: action 1; A2: action 2; O1: outcome 1; O2: outcome 2; FR1: fixed ratio 1; RR: random ratio. Data provided in *Figure 1—source data 1*.

The online version of this article includes the following source data and figure supplement(s) for figure 1:

**Source data 1.** Source files for the quantification of dopamine beta hydroxylase (DβH)-positive fibres (ventral orbitofrontal cortex [VO], lateral orbitofrontal cortex [LO], and Area 32 [A32]) and behavioural data for rats injected with saporin and inactive saporin.

**Figure supplement 1.** Whole-brain (2.5×) and zoomed-in (20×) photomicrographs showing the volume of dopamine beta hydroxylase (DβH)-positive fibres (%) in the ventral orbitofrontal cortex (VO), lateral orbitofrontal cortex (LO), and Area 32 (A32) of a representative control (CTL) animal and of two saporin (SAP)-treated rats, one with the maximum (MAX) depletion and one with the minimum (MIN) depletion.

**Figure supplement 2.** Quantification of dopamine beta hydroxylase (DβH)-positive fibres in other prefrontal cortex regions.

**Figure supplement 2—source data 1.** Source files for the quantification of dopamine beta hydroxylase (DβH)-positive fibres (medial orbitofrontal cortex [MO], secondary motor cortex [M2], and insula) for rats injected with saporin and inactive saporin.

**Figure supplement 3.** Consumption tests performed immediately after the initial (**A**) and reversal (**B**) instrumental tests (Ndev: non-devalued; Dev: devalued).

**Figure supplement 3—source data 1.** Source files for the consumption tests for rats injected with saporin and inactive saporin.

---

no significant interactions (all $F_{(1,53)}$ values < 3.7, p-values > 0.05). All groups also showed sensitivity to the change in outcome value during the outcome devaluation test, indicating that rats learned the A-O associations and the current value of the outcomes, that is, goal-directed behaviour was intact (*Figure 1E*). Indeed, we found a significant effect of devaluation (Ndev vs. Dev; $F_{(1,53)}$ = 79.62, p<0.001), but no effect of group (Pre vs. Post; $F_{(1,53)}$ = 0.21, p=0.65) or treatment (CTL vs. SAP; $F_{(1,53)}$ = 0.15, p=0.70), and no significant interactions between these factors (all $F_{(1,53)}$ values <1.78, p-values >0.18). In addition, when given concurrent access to both outcomes, all groups consumed more of the non-devalued outcome, thereby demonstrating the efficacy of the satiety-induced outcome devaluation procedure (*Figure 1—figure supplement 3A*). Thus, NA depletion in PFC regions does not appear to affect the initial learning or expression of goal-directed actions.

## NA signalling in the OFC is required to adapt to changes in outcome identity

Next, we tested whether NA depletion affected the ability to encode and express new A-O associations using an instrumental outcome-identity reversal paradigm. We found that performance increased across reversal training days ($F_{(1,53)}$ = 22.73, p<0.001), but there was no effect of group ($F_{(1,53)}$ = 0.81, p=0.37), treatment ($F_{(1,53)}$ = 0.08, p=0.78), or any significant interactions between these factors (all $F_{(1,53)}$ values < 1.86, p-values > 0.17) (*Figure 1F*). We then evaluated if the rats were able to use these reversed associations in the outcome devaluation test (*Figure 1G*). While the CTL groups reduced their responding on the action associated with the devalued outcome, NA-depleted animals did not. Statistical analyses confirmed this observation revealing no main effects of group, treatment, or devaluation (all $F_{(1,53)}$ values <2.89, p-values >0.10), and no significant devaluation × group interaction ($F_{(1,53)}$ = 0.007, p=0.93). However, there was a significant devaluation × treatment interaction ($F_{(1,53)}$ = 5.00, p<0.05) and simple effect analyses confirmed that the CTL groups biased their responding toward the lever associated with the non-devalued outcome

$(F_{(1,53)} = 7.35, p<0.01)$, while rats in the depleted groups did not $(F_{(1,53)} = 0.15, p=0.70)$. Importantly, all groups rejected the devalued food during the consumption test (*Figure 1—figure supplement 3B*). These results show that depletion of NA innervation to the OFC and other prefrontal regions renders rats unable to associate new outcomes to acquired actions. Importantly, this deficit was present in rats that received NA depletion before (group Pre) and after (group Post) learning the initial A-O associations.

## DA signalling in the OFC is not required to adapt to changes in outcome identity

Next, we examined whether DA signalling in the OFC is required for updating goal-directed actions. We first assessed the impact of full catecholaminergic (CA) depletion on outcome-identity reversal by infusing the CA-targeting toxin 6-OHDA in the OFC (6-OHDA n=12; CTL n=8). We then depleted DA afferents by combining the 6-OHDA infusion with a systemic injection of desipramine (Desi) to protect NA fibres (6-OHDA+Desi, n=9; CTL n=8). The quantification of NA and DA afferents was assessed in VO, LO, A32d, and A32v, as shown in *Figure 2* (for additional photomicrographs, see *Figure 2—figure supplement 1*). Depletion of CA afferents was assessed via TH immunostaining (*Figure 2B*). In the OFC, we found a significant within-subject effect of region (VO vs. LO; $F_{(1,26)} = 8.17$, p<0.01) and fewer TH-positive fibres in the 6-OHDA group (n=12), as compared to the CTL (n=8) and 6-OHDA+Desi (n=9) groups $(F_{(1,26)} = 41.14, p<0.001)$, which also differed $(F_{(1,26)} = 15.7, p<0.01)$. There was also a significant interaction between these factors $(F_{(1,26)} = 15.17, p<0.01)$ and simple effects revealed fewer immunoreactive TH fibres in the VO vs. the LO for the 6-OHDA+Desi group $(F_{(1,26)} = 20.66, p<0.001)$, but not for the CTL $(F_{(1,26)} = 1.14, p=0.3)$ or the 6-OHDA groups $(F_{(1,26)} = 2.74, p=0.11)$. Importantly, significantly fewer TH-positive fibres were also present in A32 for the 6-OHDA group $(F_{(1,26)} = 12.24, p<0.01)$, as compared to the CTL and the 6-OHDA+Desi groups, which did not differ $(F_{(1,26)} = 0.56, p=0.46)$ (*Figure 2B*).

The quantification of DβH-positive fibres is shown in *Figure 2D* (for additional photomicrographs, see *Figure 2—figure supplement 1*). In the OFC, main effects analyses revealed a significant within-subject effect of region (VO vs. LO; $F_{(1,26)} = 53.25, p<0.001$) and fewer DβH-positive fibres in the 6-OHDA group (n=12), as compared to the CTL (n=8) and the 6-OHDA+Desi groups (n=9) $(F_{(1,26)} = 511.47, p<0.001)$, which did not differ $(F_{(1,26)} = 3.24, p=0.08)$. There was also a region × treatment interaction $(F_{(1,26)} = 9.52, p<0.01)$ and simple effect analyses revealed that 6-OHDA induced a significant reduction in the volume of NA fibres in both the VO $(F_{(1,26)} = 400.75, p<0.001)$ and the LO $(F_{(1,26)} = 459.37, p<0.001)$. Significantly fewer DβH-positive fibres were also observed for the 6-OHDA+Desi group in the LO $(F_{(1,26)} = 6.41, p<0.05)$, but not in the VO $(F_{(1,26)} = 0.83, p=0.37)$. A significant reduction in NA innervation was also observed in A32 for the 6-OHDA group compared to the CTL and the 6-OHDA+Desi groups $(F_{(1,26)} = 137.7, p<0.001)$, which did not differ $(F_{(1,26)} = 0.05, p<0.001)$. This further indicates that our depletion was not specific to the OFC (see also *Figure 1C*). As shown in *Figure 2—figure supplement 2* 6-OHDA lesions also led to a significant decrease in NA innervation in MO $(F_{(1,25)} = 79.9, p<0.001)$ and M2 $(F_{(1,27)} = 21.08, p<0.001)$, but not in the insula cortex $(F_{(1,23)} = 1.81, p=0.19)$. The volume of DβH fibres did not differ between the CTL and the 6-OHDA+Desi groups in any of these regions (largest F=1.12, p=0.3).

All rats first underwent instrumental training and an initial outcome-specific devaluation test (see *Figure 3—figure supplement 1* and *Figure 3—figure supplement 2* for behavioural results from this initial phase). Following surgery and recovery, the animals were then trained with reversed instrumental associations. We found that rats with 6-OHDA infusions (n=12) responded more during reversal training than the CTL (n=8; $F_{(1,18)} = 14.74, p<0.01$) (*Figure 3B*). There was also an overall increase in response rate $(F_{(1,18)} = 44.69, p<0.001)$ across training and a significant group × day interaction $(F_{(1,18)} = 7.25, p<0.05)$, indicating that this increase in responding was greater for the 6-OHDA group than for CTL.

During the outcome devaluation test following reversal training (*Figure 3C*), CTL rats successfully adjusted their behaviour according to the new A-O contingencies, but the 6-OHDA group did not. Statistical analysis revealed no main effect of group $(F_{(1,18)} < 0.001, p>0.975)$, but a main effect of devaluation $(F_{(1,18)} = 8.72, p<0.01)$ and a group × devaluation interaction that approached statistical significance $(F_{(1,18)} = 3.90, p=0.06)$. Simple effect analyses confirmed that CTL rats responded more on the lever associated with the devalued outcome $(F_{(1,18)} = 10.12, p<0.01)$, while 6-OHDA rats did not

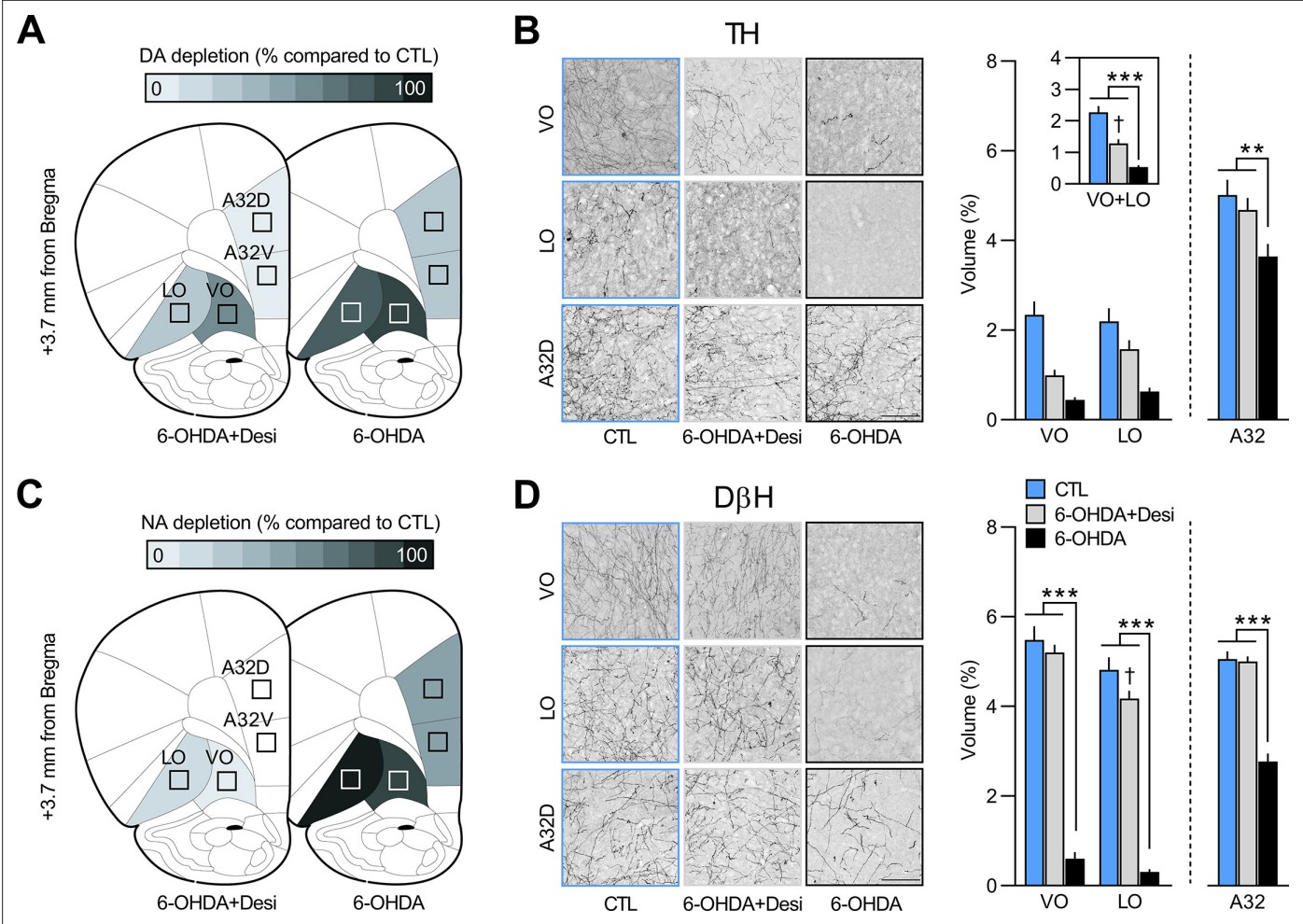

**Figure 2.** Regions where tyrosine hydroxylase (TH, panel **A**) and dopamine beta hydroxylase (DβH, panel **C**) positive fibres were quantified (Area 32 dorsal: A32d; Area 32 ventral: A32v; ventral orbitofrontal cortex: VO; lateral orbitofrontal cortex: LO); schematic adapted from Figure 9 of The Rat Brain in Stereotaxic Coordinates (*Paxinos and Watson, 2014*). Representative photomicrographs and quantification of fibres in VO, LO, and A32d for TH (**B**) and DβH immunostaining (**D**) in each of the three groups. Data are presented as mean + SEM. **$p < 0.01$, ***$p < 0.001$, †$p < 0.05$ vs. CTL group. Scale bars: 100 μm. 6-OHDA: 6-hydroxydopamine; 6-OHDA+Desi: 6-hydroxydopamine+desipramine; CTL: control. Data provided in *Figure 2—source data 1*.

The online version of this article includes the following source data and figure supplement(s) for figure 2:

**Source data 1.** Source files for the quantification of tyrosine hydroxylase (TH)-positive and dopamine beta hydroxylase (DβH)-positive fibres (ventral orbitofrontal cortex [VO], lateral orbitofrontal cortex [LO], and Area 32 [A32]) for rats injected with 6-hydroxydopamine (6-OHDA) or 6-OHDA+desipramine.

**Figure supplement 1.** Whole-brain (2.5×) and zoomed-in (20×) photomicrographs showing the volume of dopamine beta hydroxylase (DβH)-positive fibres (%) in the ventral orbitofrontal cortex (VO), lateral orbitofrontal cortex (LO), and Area 32 (A32) of a representative control (CTL), 6-OHDA+desipramine (6-OHDA+Desi), and 6-OHDA animal.

**Figure supplement 2.** Quantification of dopamine beta hydroxylase (DβH)-positive fibres in other prefrontal cortex regions.

**Figure supplement 2—source data 1.** Source files for the quantification of tyrosine hydroxylase (TH)-positive and dopamine beta hydroxylase (DβH)-positive fibres (medial orbitofrontal cortex [MO], secondary motor cortex [M2], and insula) for rats injected with 6-hydroxydopamine (6-OHDA) or 6-OHDA+desipramine.

show this preference ($F_{(1,18)} = 0.6$, $p=0.45$). Importantly, when given access to both rewards, all groups consumed more of the non-devalued than of the devalued food (*Figure 3—figure supplement 1D*).

*Figure 3D* shows responding during reversal training for the 6-OHDA+Desi (n=9) and the CTL (n=8) groups. Lever pressing increased across reversal training days ($F_{(1,15)} = 15.76$, $p<0.01$) and there was no main effect of group ($F_{(1,15)} = 0.52$, $p=0.48$) or group × day interaction ($F_{(1,15)} = 0.15$, $p=0.70$). Moreover,

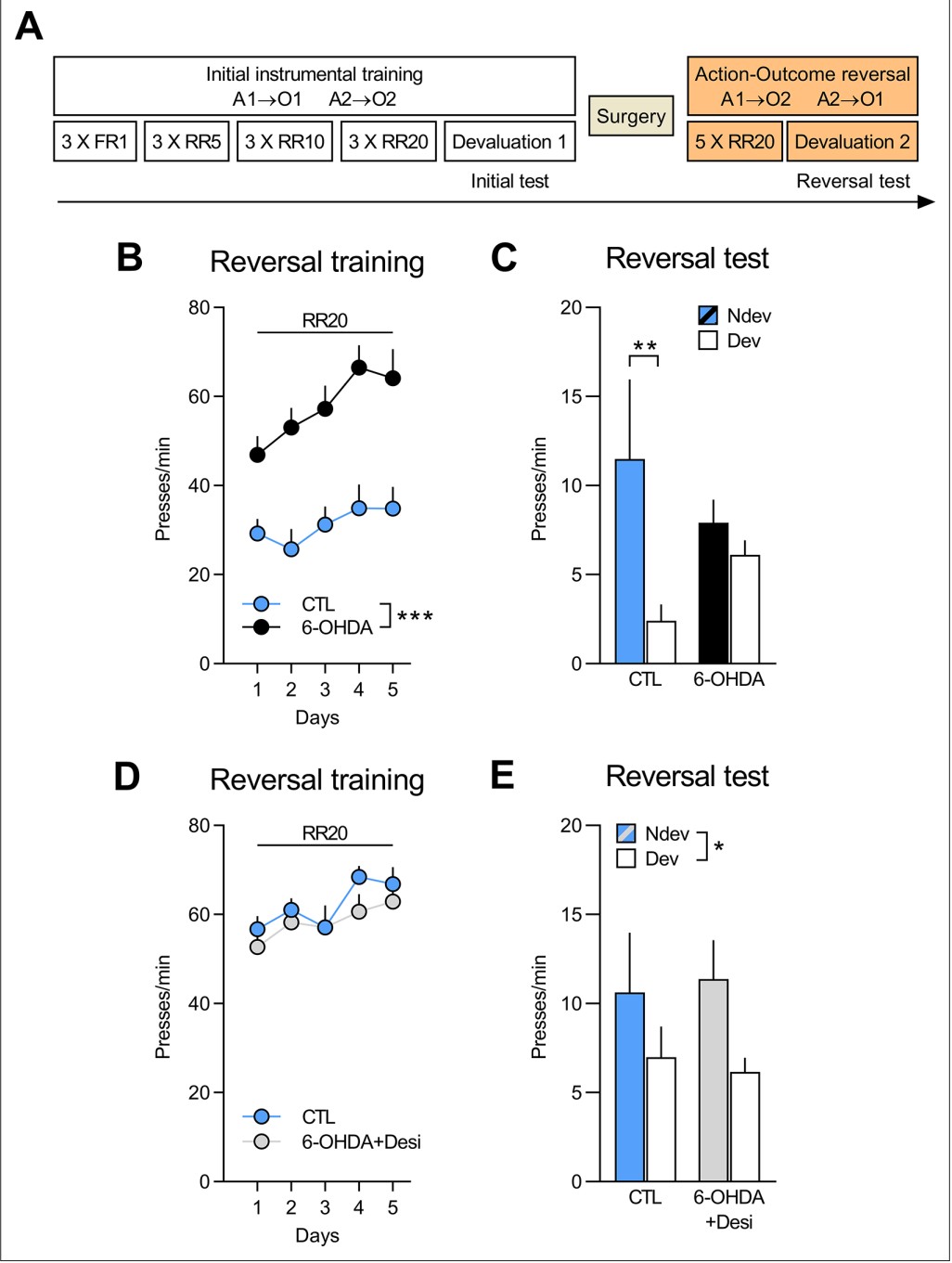

**Figure 3.** NA, but not DA, signalling in the OFC is required to adapt to changes in outcome identity. (**A**) Experimental timeline. After the initial instrumental training and outcome devaluation testing, rats were injected in the orbitofrontal cortices (OFC) with either vehicle (CTL n=8; CTL n=8), 6-OHDA coupled with desipramine (to specifically target DA neurons, 6-OHDA+Desi n=9) or 6-OHDA alone (to target all catecholaminergic [CA] neurons, n=12). (**B, D**) Rate of lever pressing across reversal training, data is presented collapsed across the two actions. (**C, E**) Reversal instrumental test in extinction following satiety-induced devaluation (Ndev: non-devalued; Dev: devalued). Data are presented as mean + SEM. *p<0.05, **p<0.01, ***p<0.001. 6-OHDA: 6-hydroxydopamine; 6-OHDA+Desi: 6-hydroxydopamine+desipramine; CTL: control; A1: action 1; A2: action 2; O1: outcome 1; O2: outcome 2; FR1: fixed ratio 1; RR: random ratio. Data provided in *Figure 3—source data 1*.

The online version of this article includes the following source data and figure supplement(s) for figure 3:

**Source data 1.** Source files for the behavioural data from the reversal phase for rats injected with

*Figure 3 continued*

6-hydroxydopamine (6-OHDA) or 6-OHDA+desipramine.

**Figure supplement 1.** Initial training and test for rats to be injected with 6-OHDA (n=12) and control rats (CTL; n=8).

**Figure supplement 1—source data 1.** Source files for the behavioural data from the initial phase for rats injected with 6-hydroxydopamine (6-OHDA).

**Figure supplement 2.** Initial training and test for rats to be injected with 6-OHDA+Desi (n=9) and control rats (CTL; n=8).

**Figure supplement 2—source data 1.** Source files for the behavioural data from the initial phase for rats injected with 6-hydroxydopamine (6-OHDA)+desipramine.

---

we found that both groups showed goal-directed behaviour and biased their choice towards the action associated with the non-devalued outcome (*Figure 3E*). Statistical analyses confirmed a main effect of devaluation ($F_{(1,15)}$ = 5.25, $p<0.05$), but no main effect of group ($F_{(1,15)}$ < 0.001, $p>0.975$) or group × devaluation interaction ($F_{(1,15)}$ = 0.17, $p=0.69$). Both groups also consumed more of the non-devalued food during the consumption test (*Figure 3—figure supplement 2D*).

These results further support a role for NA in updating previously learned goal-directed actions. We show that full CA depletion (DA + NA) in the OFC, A32, and M2 impairs performance in an outcome-identity reversal task, while depletion restricted to DA innervation leaves performance intact.

## Selective expression of inhibitory DREADDs in LC: $^{vlOFC}$ or LC:$^{A32}$ NA projections

In the previous two approaches, we used pharmacologic ablation to target NA signalling in the OFC. However, injection of anti-DHβ SAP or 6-OHDA in the OFC also caused a significant reduction of fibres in other regions of the PFC, most likely because of NA fibres crossing the OFC before entering these other prefrontal regions (*Chandler and Waterhouse, 2012*). Most notably, in both of the previous experiments, we observed depletion of NA fibres in A32 (or prelimbic cortex), a region that has been heavily implicated in goal-directed behaviour (*Corbit and Balleine, 2003*; *Killcross and Coutureau, 2003*; *Tran-Tu-Yen et al., 2009*). As such, while we were able to demonstrate that NA, but not DA, signalling in the PFC is necessary to adapt to changes in outcome identity, we could not conclusively attribute our behavioural effects to NA depletion in the OFC and not in A32. Moreover, given that our approach involved permanent lesions of NA fibres, we were unable to ascertain if NA signalling was required to encode and/or recall the new A-O associations.

Therefore, to address the regional and temporal specificity of the behavioural effect, we generated CAV2-PRS-hM4Di-mCherry, a canine adenoviral vector containing PRS, an NA-specific promoter, driving an HA-tagged hM4Di, an inhibitory DREADDs, and an mCherry expression cassette (*Figure 4A*). The validation of the construct is described in the Methods section and the corresponding results are shown in *Figure 4—figure supplement 1*. CAV2 vectors are readily taken up at presynapse and trafficked via retrograde transport to the soma of projecting neurons. CAV2-PRS-hM4Di-mCherry was infused in either the OFC or A32 to target either LC:$^{vlOFC}$ or LC:$^{A32}$ NA projections. *Figure 4C* shows retrograde transport of the vector and mCherry in NA cells of the LC following injection of CAV2-PRS-hM4Di-mCherry in the OFC. *Figure 4D* shows the colocalization of mCherry and HA immunoreactivity in the LC, indicative of a selective expression of HA-hM4Di. As expected, while mCherry staining is present at injection sites, reflecting local cortico-cortical connections that are not NA dependent (*Figure 4B*), HA-immunoreactive cell bodies were found exclusively in the LC (*Figure 4D*). These data are consistent with NA-specific expression of the HA-tagged hM4Di due to PRS, and nonselective expression of mCherry which is under the control of hSyn.

## Silencing of LC:$^{vlOFC}$, but not LC:$^{A32}$, projections impairs adaptation to changes in the A-O association

Rats received bilateral injections of CAV2-PRS-hM4Di-mCherry in either the OFC (n=25) or the A32 (n=17). Rats were then trained and tested as shown in *Figure 5A*. Following the initial instrumental training and outcome devaluation testing (*Figure 5—figure supplement 1A–C* for LC:$^{vlOFC}$ and *Figure 5—figure supplement 2A–C* for LC:$^{A32}$), both groups were further divided into two according

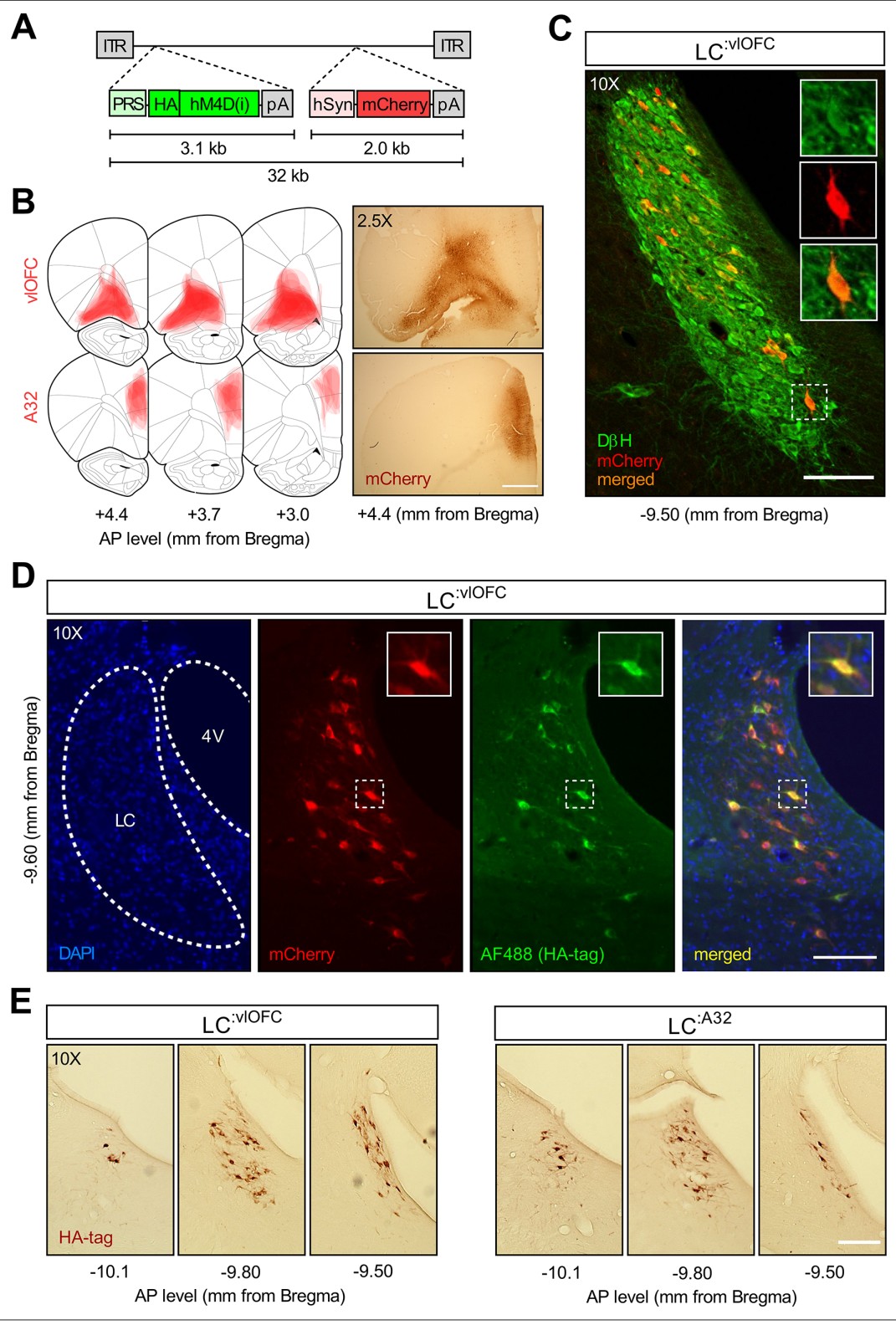

**Figure 4.** Histological examination following CAV2-PRS-hM4Di-mCherry injections.Validation of the CAV2-PRS-hM4Di-mCherry construct. (**A**) CAV2-PRS-hM4Di-mCherry, a vector bearing a noradrenergic (NA)-specific promoter (PRS) that drives the expression of inhibitory DREADDs tagged with HA (hM4Di), and an mCherry expression cassette under the neuronal-specific promoter hSyn. (**B**) Extent of viral expression across subjects and representative whole-brain photomicrographs (2.5×) showing injection sites stained for mCherry. Schematics

*Figure 4 continued on next page*

*Figure 4 continued*

adapted from Figures 8, 9 and 11 of The Rat Brain in Stereotaxic Coordinates (**Paxinos and Watson, 2014**). (**C**) Immunofluorescent staining for dopamine beta hydroxylase (DβH) and mCherry in the locus coeruleus (LC) of a representative rat injected with CAV2-PRS-hM4Di-mCherry in the orbitofrontal cortices (OFC). (**D**) High colocalization of immunofluorescent staining for HA (tag of inhibitory DREADDs) and mCherry in the LC of the same representative rat injected in the OFC. (**D**) Comparison of antero-posterior DAB staining for HA in two representative rats, one injected in the OFC, the other in A32. Scale bar panel B: 1 mm. Scale bars panels C, D, E: 100 μm.

The online version of this article includes the following source data and figure supplement(s) for figure 4:

**Figure supplement 1.** Validation of the CAV2-PRS-hM4Di-mCherry construct.

**Figure supplement 1—source data 1.** Source file for the quantification of locus coeruleus (LC) hM4Di/C-Fos-positive cells.

to their initial performance: one group that would receive vehicle (Veh; n=12 for OFC-injected rats; n=8 for A32-injected rats) and one group that would receive deschloroclozapine dihydrochloride (DCZ; n=13 for OFC-injected rats; n=9 for A32-injected rats), a high affinity and selective DREADDs agonist (**Nagai et al., 2020**; **Nentwig et al., 2021**; **Oyama et al., 2022**), during the reversal training.

A linear trend across reversal training sessions was detected for OFC-injected rats (**Figure 5B**; $F_{(1,23)}$ = 37.53, p<0.001), but not A32-injected rats (**Figure 5D**; $F_{(1,15)}$ = 0.16, p=0.69), with no difference between the Veh and the DCZ groups for either OFC- ($F_{(1,23)}$ = 0.39, p=0.54) or A32-injected rats ($F_{(1,15)}$ = 0.01, p=0.92), and no interactions between these factors (largest $F_{(1,23)}$ value = 2.82, p=0.11). All rats underwent two outcome devaluation tests, once under vehicle (-), once under DCZ (+), with the order counterbalanced. This yielded a 2 (between) × 2 (within) factorial design with four conditions of interest: vehicle during training and test (-/-), vehicle during training and DCZ during test (-/+), DCZ during training and vehicle at test (+/-), and DCZ during training and test (+/+).

For rats injected in the OFC (**Figure 5C**), we found that only the CTL group (-/-) showed goal-directed behaviour and performed the action associated with the non-devalued outcome more than the action associated with the devalued outcome. By contrast, rats with bilateral LC:$^{vlOFC}$ NA projections silenced during the reversal training (+/-), or the outcome devaluation test (-/+), or during both phases (+/+) failed to display this preference in responding. Statistical analyses confirmed no significant between- or within-subject main effects (largest $F_{(1,46)}$ value = 1.22, p=0.28), but a significant three-way interaction (devaluation × treatment during acquisition × treatment during test: $F_{(1,23)}$ = 5.45, p<0.05). Simple effects analyses confirmed that only rats that received vehicle during both training and test (-/-) biased their choice towards the lever associated with the non-devalued outcome ($F_{(1,23)}$ = 7.10, p<0.05), while rats in the -/+ ($F_{(1,23)}$ = 0.09, p=0.77), +/- ($F_{(1,23)}$ = 1.93, p=0.18), and +/+ conditions ($F_{(1,23)}$ = 0.01, p=0.92) did not.

By contrast, silencing LC:$^{A32}$ NA projections left goal-directed behaviour intact (**Figure 5E**). Indeed, we found a main effect of devaluation ($F_{(1,15)}$ = 5.76, p<0.05), but no effect of treatment during acquisition ($F_{(1,15)}$ = 0.35, p=0.56) or treatment during test ($F_{(1,15)}$ = 0.53, p=0.47), and no significant interactions between these factors (largest $F_{(1,15)}$ value = 1.62, p-values > 0.21).

Importantly, consumption tests performed immediately after the reversal tests revealed that all groups consumed more of the non-devalued outcome indicating that the satiety-induced devaluation was effective and that DCZ injections did not disrupt the rats' ability to distinguish between devalued and non-devalued rewards (**Figure 5—figure supplement 1D** and **Figure 5—figure supplement 2D** for LC: $^{vlOFC}$ and LC:$^{A32}$, respectively). Together, these results indicate that LC NA projections to the OFC, but not to A32, are required to both encode and recall changes in the identity of the expected outcome.

## Discussion

Goal-directed actions are the expression of learned associations between an action and the outcome it produces. These associations are however flexible, being amenable to updating when the identity of the outcome changes. Our data demonstrate that NA inputs to the OFC might be an essential component of this updating process. This conclusion is based on a body of complementary evidence. First, we demonstrated that animals with a loss of NA inputs in the OFC can initially learn and express A-O

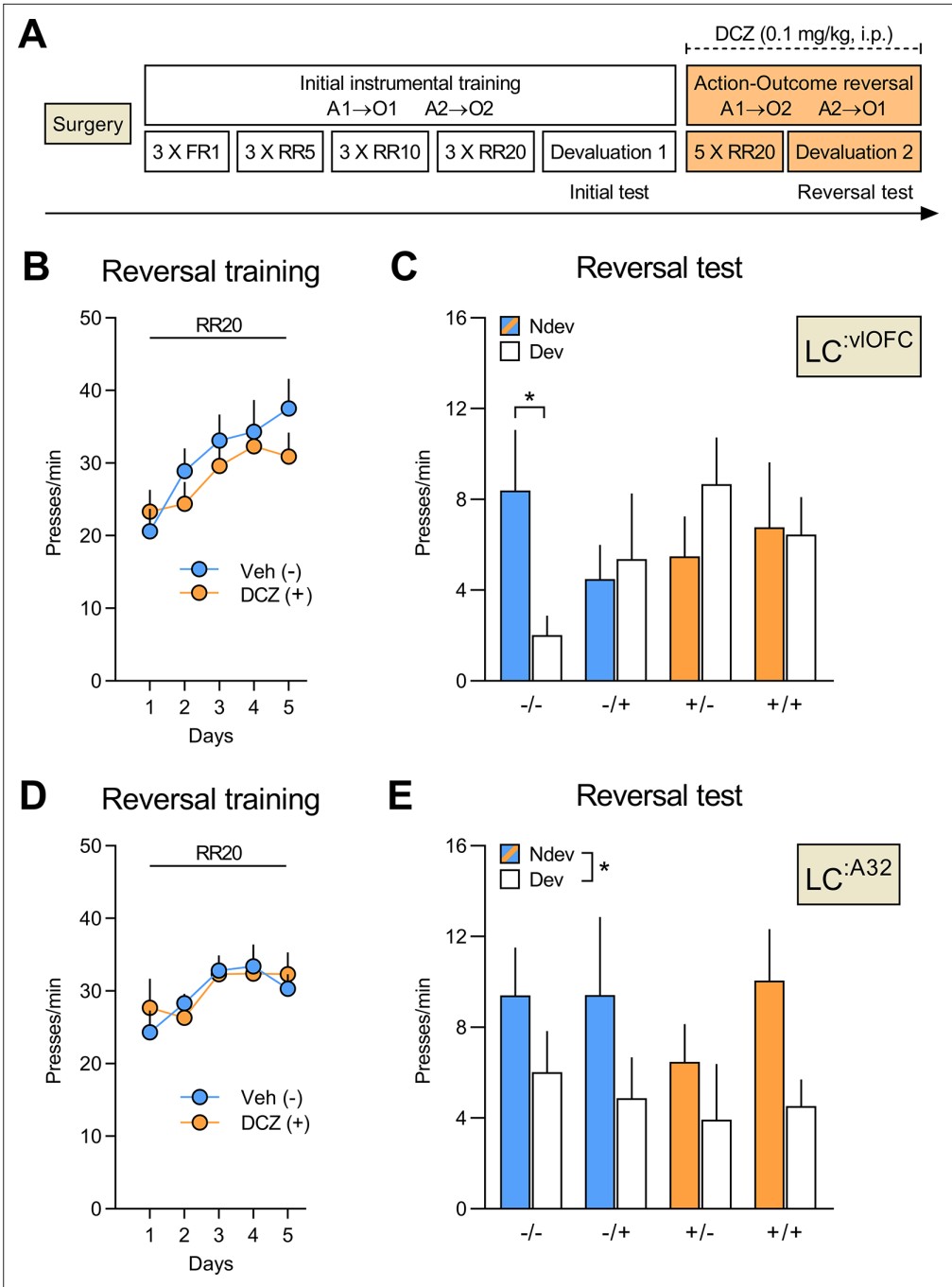

**Figure 5.** Silencing of LC:vlOFC, but not LC:A32, projections impairs adaptation to changes in the A-O association.
(**A**) Timeline for rats injected with CAV2-PRS-hM4Di-mCherry in either the orbitofrontal cortices (OFC) or Area 32
(A32). Each rat was injected with either vehicle (-) or DCZ (+) during the reversal training and then tested twice,
once under DCZ and once under vehicle, with the test order counterbalanced. (**B**) Reversal training in rats injected
in the OFC (Veh n=12; DCZ n=13), data is presented collapsed across the two actions. (**C**) Reversal instrumental
test following satiety-induced devaluation in rats injected in the OFC (Ndev: Non-devalued; Dev: devalued).
(**D**) Reversal training in rats injected in A32 (Veh n=8; DCZ n=9), data is presented collapsed across the two actions.
(**E**) Reversal instrumental test following satiety-induced devaluation in rats injected in A32. Data are presented as
mean + SEM. *p<0.05. A1: action 1; A2: action 2; O1: outcome 1; O2: outcome 2; FR1: fixed ratio 1; RR: random
ratio; DCZ: deschloroclozapine; LC: locus coeruleus. Data provided in *Figure 5—source data 1*.

The online version of this article includes the following source data and figure supplement(s) for figure 5:

**Source data 1.** Source file for the behavioural data from the reversal phase for LC:vlOFC and LC:A32 rats.

*Figure 5 continued on next page*

*Figure 5 continued*

**Figure supplement 1.** Initial training and test for rats injected with CAV2-PRS-hM4Di-mCherry in the orbitofrontal cortex (OFC).

**Figure supplement 1—source data 1.** Source file for the behavioural data from the initial phase for LC:$^{vlOFC}$ rats.

**Figure supplement 2.** Initial training and test for rats injected with CAV2-PRS-hM4Di-mCherry in area 32 (A32).

**Figure supplement 2—source data 1.** Source file for the behavioural data from the initial phase for LC:$^{A32}$ rats.

contingencies, but are impaired when the identity of the outcome has been modified. Importantly, such deficits were also observed when NA depletion occurred immediately before the encoding of the new A-O contingencies. We then showed that this impairment was selective to NA inputs, because combined depletions of DA and NA, but not of DA alone, induced a profound deficit in outcome reversal. Finally, we investigated the temporal and anatomical specificity of this effect using an NA-specific retrograde virus carrying inhibitory DREADDs to selectively target either LC:$^{vlOFC}$ or LC:$^{A32}$ pathways. We found that silencing LC:$^{vlOFC}$, but not LC:$^{A32}$, projections impaired the rats' ability to acquire and express the reversed instrumental contingencies.

## NA inputs into the OFC, but not the A32, are required for A-O updating

The use of the SAP toxin led to a dramatic decrease of NA fibres density in all analysed cortical areas (*Figure 1B* and *Figure 1—figure supplement 2A*). This may be due to diffusion of the toxin from the injection site or to the existence of collateral LC neurons and/or fibres passing through the ventral portion of the OFC, but targeting other cortical areas (*Cerpa et al., 2019*). However, injection of 6-OHDA led to less offsite NA depletion suggesting that a large part of the previous observation is toxin-specific. Indeed, no significant loss of NA fibres was visible in the insula cortex (*Figure 2—figure supplement 2B*), which has been previously implicated in goal-directed behaviour (*Balleine and Dickinson, 2000*; *Parkes and Balleine, 2013*; *Parkes et al., 2015*). We did nevertheless observe an offsite depletion in more proximal prefrontal areas (prelimbic/A32 and MO), albeit a more modest depletion that what was observed using the SAP toxin. Several studies have described the projection pattern of LC cells. These studies, using various techniques, indicate that LC cells mainly target a single region, and that only a small proportion of LC neurons collateralize to minor targets (*Plummer et al., 2020*; *Kebschull et al., 2016*; *Uematsu et al., 2017*; *Chandler et al., 2014*). Therefore, even if the OFC NA innervation is presumably specific (*Chandler et al., 2013*), we cannot rule out a possible collateralization of some neurons toward neighbouring prefrontal areas (including A32 and MO). We have previously discussed that the posterior ventral portion of the OFC is an entry point for LC fibres en passant, which ultimately target other prefrontal areas (*Cerpa et al., 2019*).

We then used a CAV2 vector carrying the NA-specific promoter PRS to target either the LC:$^{vlOFC}$ or the LC:$^{A32}$ pathway (*Hayat et al., 2020*; *Hirschberg et al., 2017*). It has been shown that the CAV2 vector can infect axons-of-passage, however the vector does not spread more than 200 μm from the injection site (*Schwarz et al., 2015*). Therefore, when targeting the OFC, we injected anteriorly to the level where the highest density of fibres of passage is expected (*Cerpa et al., 2019*) in order to minimize infection of such fibres and restrict inhibition to our pathway of interest.

Overall, the current behavioural results are in line with our previous work showing that the ability to associate new outcomes to previously acquired actions is impaired following chemogenetic inhibition of the VO and LO (*Parkes et al., 2018*) or disconnection of the VO and LO from the submedius thalamic nucleus (*Fresno et al., 2019*). These results point to a role for the ventral and lateral parts of the OFC and its NA innervation in updating A-O associations. However, it is worth mentioning that different subregions of the OFC, both along the medio-lateral and antero-posterior axes of OFC, display clear functional heterogeneities (*Bradfield and Hart, 2020*; *Izquierdo, 2017*; *Panayi and Killcross, 2018*; *Bradfield et al., 2018*; *Barreiros et al., 2021*). Therefore, while we have previously focused on the anatomical heterogeneity of the NA innervation in these prefrontal subregions (*Cerpa et al., 2019*), a thorough characterization of its functional role in each of these subregions still needs to be addressed. We must also acknowledge that only male rats were used in the current study. The LC displays some anatomical and physiological variations between male and female rats (*Joshi and Chandler, 2020*), therefore a thorough characterization would also need to integrate this sex factor.

Our key finding is that NA inputs to the OFC are required for updating the association between an action and its outcome. Indeed, a similar impairment was observed when NA depletion was performed either prior to initial training or prior to reversal training, which indicates that the reversal period is critically reliant on NA inputs. In addition, chemogenetic silencing of the LC:vlOFC pathway before the reversal training, or before testing also produced similar impairments, which further demonstrates that OFC NA inputs are required for both the encoding and the recall of new A-O associations. These results are consistent with recent views on the role of the OFC in goal-directed behaviour (*Parkes et al., 2018*; *Panayi and Killcross, 2018*; *Cerpa et al., 2021*).

In contrast to NA inputs to the OFC, our results show that NA inputs to A32 (the prelimbic cortex) are not required for responding based on initial or reversed instrumental contingencies. These data add to the current literature indicating a major dissociation in the role of NA inputs to different prefrontal regions (*Robbins and Arnsten, 2009*). Indeed, NA inputs to the medial wall of the PFC are required for attentional regulation. Specifically, lesioning NA inputs (*McGaughy et al., 2008*; *Newman et al., 2008*) or chemogenetic inhibition of NA inputs to the mPFC (*Cope et al., 2019*) alters attentional set-shifting, while NA recapture inhibition via atomoxetine improves it (*Newman et al., 2008*). OFC NA depletion can also alter cue-outcome reversal, but not dimensional shift, in an attentional set-shifting task (*Mokler et al., 2017*). Recently, it was also shown that reversible inactivation of the medial OFC (mOFC) decreased performance accuracy on a two-armed bandit task in rats (*Swanson et al., 2022*). Interestingly, performance accuracy was also impaired following systemic, but not intra-mOFC, administration of an NA antagonist (*Swanson et al., 2022*). This result seems consistent with our finding showing that reversal learning and expression is intact when the LC:A32 pathway is silenced and may suggest a potential role for the ventral and lateral regions of OFC, rather than mOFC, in this effect.

## NA, but not DA, inputs to the OFC are required for A-O updating

Using a strategy which allows for a differential depletion of DA and/or NA fibres, we found that NA-dependent mechanisms are required during the encoding and recall of new A-O. The role of cortical DA-dependent mechanisms in goal-directed behaviour remains poorly understood, but we have previously shown that DA signalling in the prelimbic cortex/A32 plays a critical role in the detection of contingency degradation (*Naneix et al., 2009*). Such detection is likely to involve the processing of non-expected rewards which induces, at the level of A32, a DA-dependent reward prediction error signal (*Montague et al., 2004*; *Schultz and Dickinson, 2000*). These results therefore raise the possibility that the coordination of goal-directed behaviour under environmental changes might depend on a DA-A32 system to adapt to causal contingencies and an NA-OFC system to adapt to changes in outcome identity (*Cerpa et al., 2021*). However, it is not yet clear if the NA-OFC system is also involved in detecting the causal relationship between an action and its outcome (see *Cerpa et al., 2021*, for a discussion). Some have reported impaired adaptation to contingency changes following inhibition of VO and LO or BDNF knockdown in these regions (*Whyte et al., 2021*; *Zimmermann et al., 2017*), while another study showed that inhibition of VO/LO leaves sensitivity to degradation intact, at least during an initial test (*Zimmermann et al., 2018*). Interestingly, a recent paper in marmosets demonstrates that inactivation of anterior OFC (Area 11) improves instrumental contingency degradation, whereas overactivation impairs degradation (*Duan et al., 2021*). The potential role of the rodent ventral and lateral regions of OFC, and of NA innervation to the OFC, in adapting to degradation of instrumental contingencies requires further investigation.

## Updating goal-directed behaviour

When trained on reversed contingencies, animals encode the new A-O associations (*Fresno et al., 2019*; *Parkes et al., 2018*). Under similar experimental conditions, past research has shown that reversal learning performance is the result of updating prior existing A-O contingencies without unlearning the initial contingencies (*Bradfield and Balleine, 2017*). In other words, the animals build a partition between a state for the new contingencies and the initial state of old contingencies (*Hart and Balleine, 2016*). Current research has proposed that the OFC is critically involved in this partition of information when task states change without explicit notice (*McDannald et al., 2011*; *Wilson et al., 2014*; *Sadacca et al., 2017*, *Wikenheiser and Schoenbaum, 2016*). Consistent with this view, chemogenetic inhibition of the OFC (ventral and lateral) impairs goal-directed responding following identity reversal (*Parkes et al., 2018*; *Howard and Kahnt, 2021*). Here, we found a similar deficit following

lesion of NA inputs to the OFC. Given that the deficit in goal-directed behaviour was restricted to the reversal phase, including both reversal training and the test based on reversed contingencies, it is likely that NA-OFC is involved in both creating new states and in the 'online' use of the information included in this new state. Such a proposal is in accordance with popular LC-NA system theories suggesting that a rise in NA activity allows for behavioural flexibility when a change in contingencies is detected (*Aston-Jones et al., 1997*; *Bouret and Sara, 2005*; *Sadacca et al., 2017*).

## Conclusion

Our results provide evidence for the involvement of NA inputs to ventral and lateral OFC in the updating and use of new A-O associations. Recent research has revealed a remarkable parcellation of cortical functions in goal-directed action (*Fresno et al., 2019*; *Turner and Parkes, 2020*; *Dalton et al., 2016*). The current study provides a clear basis for an in-depth understanding of the cortical coordination involved in executive functions.

## Methods

**Key resources table**

| Reagent type (species) or resource | Designation | Source or reference | Identifiers | Additional information |
|---|---|---|---|---|
| Antibody | Anti-DβH (mouse monoclonal) | Merck Millipore | Cat# MAB308 | 1:1000 |
| Antibody | Anti-TH (mouse monoclonal) | Merck Millipore | Cat# MAB318 | 1:2000 |
| Antibody | Anti-RFP (rabbit polyclonal) | MBL International Corporation | Cat# PM005 | 1:2000 |
| Antibody | Anti-HA (rabbit monoclonal) | Cell Signaling Technology | Cat# C29F4 (#3724) | 1:1000 |
| Antibody | Anti-c-Fos (rabbit monoclonal) | Cell Signaling Technology | Cat# 9F6 (#2250) | 1:1000 |
| Antibody | Anti-mouse biotin-conjugated (goat polyclonal) | Jackson ImmunoResearch | Cat# 115-065-062 | 1:1000 |
| Antibody | Anti-rabbit biotin-conjugated (goat polyclonal) | Jackson ImmunoResearch | Cat# 111-065-003 | 1:1000 |
| Antibody | Anti-mouse FITC-conjugated (goat polyclonal) | Jackson ImmunoResearch | Cat# 115-095-003 | 1:400 |
| Antibody | Anti-rabbit TRITC-conjugated (goat polyclonal) | Jackson ImmunoResearch | Cat# 111-025-003 | 1:200 |
| Antibody | Anti-rabbit AF488-conjugated (goat polyclonal) | Jackson ImmunoResearch | Cat# 111-545-003 | 1:1000 |
| Chemical compound, drug | Streptavidin-Alexa 488 | Thermo Fisher Scientific | Cat# S11223 | 1:500 |
| Chemical compound, drug | Anti-DβH saporin | Advanced Targeting Solutions | Cat# KIT-03 | |
| Chemical compound, drug | Mouse IgG saporin (inactive) | Advanced Targeting Solutions | Cat# IT-18 sold as KIT-03 | |
| Chemical compound, drug | Diaminobenzidine (DAB) | Sigma-Aldrich | Cat# D5905 | |
| Chemical compound, drug | Deschloroclozapine (DCZ) | MedChemExpress | Cat# HY-42110 | Injectable volume 0.1 mg/kg |
| Chemical compound, drug | 6-OHDA hydrochloride | Sigma-Aldrich | Cat# H4381 | |

*Continued on next page*

*Continued*

| Reagent type (species) or resource | Designation | Source or reference | Identifiers | Additional information |
|---|---|---|---|---|
| Chemical compound, drug | Desipramine | Sigma-Aldrich | Cat# D3900 | |
| Commercial assay or kit | Avidin-biotin-peroxydase (ABC kit) | Thermo Fisher Scientific | Cat# 32020 | |
| Transfected construct (human) | CAV2 PRS HA-hM4Di E1 hSyn mCherry E3 | https://plateau-igmm.pvm.cnrs.fr/?vector=cav-prs-ha-hm4di | | Titre $3.5 \times 10^{12}$ pp/mL |

## Animals and housing

A total of 136 male Long-Evans rats, aged 2–3 months, were obtained from the Centre d'Elevage Janvier (France). Rats were housed in pairs with ad libitum access to water and standard lab chow prior to behavioural experiments. Rats were handled daily for 3 days prior to the beginning of the experiments and were put on food restriction 2 days before behaviour to maintain them at approximately 90% of their ad libitum feeding weight. The facility was maintained at 21 ± 1°C on a 12 hr light/dark cycle (lights on at 8:00 am). Environmental enrichment was provided by tinted polycarbonate tubes and nesting material, in accordance with current French (Council directive 2013-118, February 1, 2013) and European (directive 2010-63, September 22, 2010, European Community) laws and policies regarding animal experimentation. The experiments received approval from the local Bordeaux Ethics Committee (CE50).

## Stereotaxic surgery

For all experiments, rats were anaesthetized with 5% inhalant isoflurane gas with oxygen and placed in a stereotaxic frame with atraumatic ear bars (Kopf Instruments) in a flat skull position. Anaesthesia was maintained with 1.5% isoflurane and complemented with a subcutaneous injection of ropivacaïne (a bolus of 0.1 mL at 2 mg/mL) at the incision site. After each injection, the injector was kept in place for an additional 10 min before being removed. Rats were given 4 weeks to recover following surgery. Injection sites were confirmed histologically after the completion of behavioural experiments.

In the first experiment (n=57), we used a toxin selective for NA neurons (SAP) to target and deplete NA terminals in the VO and LO. For half of the rats ('Pre' groups, n=29), surgery was performed before the initial instrumental training and testing phase, for the other half surgery was performed after the initial training and testing ('Post' groups, n=28). Intracerebral injections were made using repeated pressure pulses delivered via a glass micropipette connected to a pressure injector (Picospritzer III, Parker). For SAP groups (Pre n=15; Post n=15), 0.1 μL of anti-DβH SAP (0.1 μg/μL) was bilaterally injected at one site targeting both VO and LO, while CTL rats (Pre n=14; Post n=13) received 0.1 μL of inactive IgG SAP (0.1 μg/μL). Injection coordinates (in mm from Bregma) were determined from the atlas of *Paxinos and Watson, 2014*: +3.5 antero-posterior (AP), ±2.2 medio-lateral (ML), and –5.4 dorso-ventral (DV).

We then used a toxin selective for CA neurons (6-OHDA hydrochloride) and a noradrenaline uptake-blocker (Desi) to target and deplete DA neurons in the VO and LO. All rats underwent surgery after the initial instrumental training phase. Rats were then allocated to the full CA depletion condition (group 6-OHDA n=12; CTL n=8) or the specific DA depletion condition (6-OHDA+Desi n=9; CTL n=8). 6-OHDA (4 μg/μL) was dissolved in vehicle solution containing 0.9% NaCl and 0.1% ascorbic acid. A volume of 0.2 μL of 6-OHDA was bilaterally injected in the OFC at the same coordinates as for the first experiment. Animals in the CTL group received injections of the vehicle solution. Thirty minutes before the surgical procedure, animals in the 6-OHDA+Desi group received a systemic (i.p.) injection of Desi (25 mg/mL) at a volume of 1 mL/kg.

In the chemogenetic experiments (n=42), we employed a canine adenovirus type 2 (CAV2) vector equipped with an NA-selective synthetic promoter (PRS) and inhibitory DREADDs (hM4Di) to

specifically target LC:$^{vlOFC}$ and LC:$^{A32}$ NA projections. This viral construct was designed and tested at the Plateforme de Vectorologie de Montpellier, Institute of Molecular Genetics, Montpellier, https://plateau-igmm.pvm.cnrs.fr/?vector=cav-prs-ha-hm4di. All rats underwent surgery before the initial instrumental training phase. All animals received 1 µL bilateral injections of the adenovirus (titre 3.5×10$^{12}$ pp/mL), which were performed using a 10 µL Hamilton syringe and a stereotax-mounted injection pump (World Precision Instruments) at a flow rate of 100 nL/min. To target both VO and LO, rats (n=25) were injected at the following coordinates (mm from Bregma): +3.7 AP, ±2.0 ML, –5.0 DV, and +3.2 AP, ±2.8 ML, –5.2 DV (2 injection sites per hemisphere). To target A32 (prelimbic cortex), rats (n=17) were injected at the following coordinates (mm from Bregma): +3.2 AP, ±0.6 ML, –3.6 DV (1 injection site per hemisphere).

## Validation of the CAV2-PRS-hM4Di-mCherry construct

Although previous studies have validated the use of the CAV2-PRS construct using a range of actuators, including excitatory and inhibitory opsins, as well as potassium channels (*Howorth et al., 2009a*; *Howorth et al., 2009b*; *Hickey et al., 2014*; *Li et al., 2016*), we used a separate cohort of rats to verify the functional effect of our DREADDs construct in vivo by quantifying changes in the expression of c-Fos, a recognized marker of neuronal activation, upon administration of DCZ. In order to obtain a high baseline of c-Fos activation, rats underwent a stress procedure. As shown in *Figure 4—figure supplement 1B*, rats were administered i.p. with either vehicle or DCZ (0.1 mg/kg) 45 min before being placed in a Plexiglas shock chamber equipped with stainless steel rods on the floor and a circuit generator connected to a scrambler and a timing unit. Rats received five shocks (0.5 s, 0.8 mA) randomly interspersed over 10 min (*stress* condition). As a control of c-Fos activation, we included in the experimental design animals administered with vehicle, but left untouched in their home cage (*no stress* condition). Rats were perfused 90 min after the procedure and coronal slices collected as described in the Histology section. For hM4Di/c-Fos colocalization analysis, sections were taken (see *Figure 4—figure supplement 1—source data 1*) from antero-posterior levels of the LC from –9.50 to –10.0 mm from Bregma (vehicle/no stress, n=2; vehicle-stress, n=4; DCZ-stress, n=4). Quantification of the percentage of LC hM4Di-positive cells (mCherry, red) that co-express c-Fos (Alexa 488, green) was performed by a trained observer blind to the experimental conditions.

## Behavioural apparatus

For all behavioural experiments, training and testing was conducted in eight identical operant chambers (40 cm width × 30 cm depth × 35 cm height, Imetronic, Pessac, France) individually enclosed in sound and light-resistant wooden chambers (74 × 46 × 50 cm$^3$). Each chamber was equipped with two pellet dispensers that delivered grain (Rodent Grain-Based Diet, 45 mg, Bio-Serv) or sugar (LabTab Sucrose Tablet, 45 mg, TestDiet) pellets into a food port when activated. For instrumental conditioning, two retractable levers were located on each side of the food port. Each chamber had a ventilation fan producing a background noise of 55 dB. During the session, the chamber was illuminated by four LEDs in the ceiling. Experimental events were controlled and recorded by a computer located in the room and equipped with the POLY software (Imetronic).

## Behavioural protocol
### Initial training and test

The training procedure was adapted from *Parkes et al., 2018*. On days 1–3, rats were trained to retrieve food pellets from the food port. During each daily session, 40 sugar and 40 grain pellets were delivered pseudo-randomly every 60 s, on average. Following food port training, rats received 12 daily sessions of instrumental training, during which they were required to learn initial A-O associations. During these sessions, each lever, in alternation, was presented twice for a maximum of 10 min or until 20 outcomes were earned. The inter-trial interval between lever presentations was 2.5 min (i.e., the session could last up to 50 min and the rats could obtain a maximum of 80 food pellets). The A-O associations and the order of lever presentations were counterbalanced between rats and days. During the first three sessions, lever pressing was continuously reinforced with a fixed ratio (FR) 1 schedule. Then, the probability of receiving an outcome was reduced, first with a random ratio (RR) 5 schedule (days 4–6), then with an RR10 (days 7–9,) and an RR20 schedule (days 10–12).

Outcome devaluation tests were performed 1 day after the last instrumental training session. First, to induce sensory-specific satiety (*Rolls et al., 1986*), rats received access to one of the two outcomes (20 g) for 1 hr in a set of plastic feeding cages to which they were previously habituated. Immediately after the satiety procedure, rats were returned to the operant chambers where they were given a choice test in extinction (i.e., unrewarded) with both levers available for 10 min. The devalued (sated) food was counterbalanced between rats. Following the extinction test, animals were returned to the plastic feeding cages and given a consumption test of satiety-induced devaluation, during which they received 10 min concurrent access to both types of food pellets (10 g of each). The amount consumed of each pellet type was measured to confirm that the satiety-induced devaluation was effective and that rats were able to distinguish between the sensory features of the different food pellets.

### Reversal training and test
Following the initial phase, rats were trained on reversed A-O associations with a procedure adapted from previous studies in our laboratory (*Fresno et al., 2019*; *Parkes et al., 2018*). Specifically, the identity of outcomes was switched so that rats had to update previously established A-O associations, always keeping a RR20 schedule of reinforcement. Following reversal training, outcome devaluation tests were conducted in the same manner as previously described.

## Chemogenetics
The DREADD agonist deschloroclozapine (DCZ) was dissolved in dimethyl sulfoxide (DMSO) to a final volume of 50 mg/mL, aliquoted in small tubes (50 µL) and stored at –80°C (stock solution). For behavioural experiments, our stock solution was diluted in physiological saline to a final injectable volume of 0.1 mg/kg and administered systemically (i.p.) 40–45 min prior to testing at a volume of 10 mL/kg. Fresh injectable solutions were prepared from stock aliquots on the day of the usage. DCZ was prepared and injected under low light conditions.

## Histology
At the end of all behavioural experiments, rats were injected with a lethal dose of sodium pentobarbital (Exagon Euthasol) and perfused transcardially with 60 mL of saline followed by 260 mL of 4% paraformaldehyde (PFA) in 0.1 M phosphate buffer (PB). Brains were removed and post-fixed in the same PFA 4% solution overnight and then transferred to a 0.1 M PB solution or to a 0.1 M PB with 30% saccharose solution (6-OHDA experiment). Subsequently, 40 µm coronal sections were cut using a VT1200S Vibratome (Leica Microsystems) or freezing microtome for the 6-OHDA experiment. Every fourth section was collected to form a series. DAB staining was performed for DβH (for the SAP and the 6-OHDA experiments), TH (6-OHDA experiment), HA, and mCherry (chemogenetic experiments).

Free-floating sections were first rinsed (4×5 min) in 0.1 M phosphate buffer saline (PBS) containing 0.3% Triton X-100 (PBST) and then incubated in PBST containing 0.5% (for mCherry) or 1% (for DβH and TH) hydrogen peroxide solution (H$_2$O$_2$) for 30 min in the dark. Further rinses (4×5 min) in PBST and a 1 hr incubation in blocking solution (PBST containing 3% goat serum) followed. Sections were then incubated with the primary antibody (mouse monoclonal anti-DβH, 1/1000; mouse monoclonal anti-TH, 1/2000; rabbit monoclonal anti-HA, 1/1000; rabbit polyclonal anti-RFP, 1/2000) diluted in blocking solution for 24 hr (for mCherry) or 48 hr (for DβH and TH) at 4°C. After rinses (4×5 min) in PBS (for DβH and TH) or PBST (for mCherry), sections were placed in a bath containing the secondary antibody (biotinylated goat anti-mouse, 1/1000; biotinylated goat anti-rabbit, 1/1000) diluted in PBS (for DβH and TH) or PBST containing 1% goat serum (for mCherry) for 2 hr at room temperature. Following rinses (4×5 min) in PBS (for DβH and TH) or PBST (for mCherry), they were then incubated with the avidin-biotin-peroxydase complex (1/200 in PBS for DβH and TH; 1/500 in PBST for mCherry) for 90 min in the dark at room temperature. H$_2$O$_2$ was added to the solution before the final staining with DAB was made (10 mg tablet dissolved in 50 mL of 0.1 M Tris buffer). Stained sections were finally rinsed with 0.05 M Tris buffer (2×5 min) and 0.05 M PB (2×5 min), before being collected on gelatin-coated slides using 0.05 M PB, dehydrated (with xylene for DβH and TH), mounted and cover-slipped using the Eukitt mounting medium.

Immunofluorescence staining was also performed for DβH, mCherry, and HA (chemogenetic experiments). Free-floating sections were first rinsed in PBS (4×5 min) and PBST (3×5 min), before being incubated in blocking solution for 1 hr at room temperature. Sections were then incubated with the

primary antibody (mouse monoclonal anti-DβH, 1/1000; rabbit polyclonal anti-RFP, 1/1000; rabbit monoclonal anti-HA, 1/1000) diluted in blocking solution for 24 hr at 4°C. Following rinses in PBS (4×5 min), they were then incubated with the secondary antibody (goat polyclonal anti-mouse FITC-conjugated, 1/400; goat polyclonal anti-rabbit TRITC-conjugated, 1/200; with goat polyclonal anti-rabbit, 1/1000) diluted in PBS for 2 hr in the dark at room temperature. Stained sections were finally rinsed with PSB (4×5 min), before being collected on gelatin-coated slides using 0.05 M PB, dehy-drated, mounted and cover-slipped using Fluoroshield with DAPI mounting medium.

Sequential immunofluorescence staining was performed for c-Fos and mCherry. Free-floating sections were first incubated overnight with an anti-c-Fos antibody (rabbit monoclonal anti-c-Fos, 1:1000), followed by a 2 hr incubation with the secondary antibody (biotinylated goat anti-rabbit, 1:1000) and an overnight incubation with streptavidin-Alexa 488 (1:500). Sections were then incu-bated overnight with an anti-RFP antibody (rabbit polyclonal anti-RFP, 1:1000), followed by a 2 hr 30 min incubation with the secondary antibody (goat anti-rabbit-TRITC, 1:200). Sections were washed with PBS in-between each step (3×10 min).

## Fibres loss quantification

To measure fibre density in the SAP and 6-OHDA experiments, we used the protocol described in *Cerpa et al., 2019*. We examined sections at +4.4, +3.7, and+3.0 (mm from Bregma) using a Nano-zoomer slide scanner with a 20× lens (Hamamatsu Photonics). Digital photomicrographs of regions of interest (ROI, square windows of 300×300 $\mu m^2$, 1320×1320 pixels) in each hemisphere were exam-ined under a 20× virtual lens with the NDP.view 2 freeware (Hamamatsu Photonics). Each ROI was outlined according to *Paxinos and Watson, 2014*. Quantification of DβH- and TH-positive fibres was performed using an automated method developed in the laboratory with the ImageJ software (*Cerpa et al., 2019*). Specifically, a digitized version of the photomicrograph was converted to black and white by combining blue, red, and green channels (weights 1, –0.25, and –0.25), subjected to a median filter (radius 3 pixels) in order to improve the signal-to-noise ratio, smoothed with a Gaussian filter (radius 8), and subtracted from the previous picture to isolate high spatial frequencies. Large stains were further eliminated by detecting them in a copy of the image. The picture was then subjected to a fixed threshold (grey level 11) to extract stained elements, and the relative volume occupied by fibres estimated by the proportion of detected pixels in the ROI. As a control for poor focus, the same images were analysed a second time while allowing lower spatial frequencies (Gaussian filter radius 20). The ratio between the proportions of pixels detected by the two methods was used as a criterion to eliminate blurry images.

## Experimental design and data analysis

Each rat was assigned a unique identification number that was used to conduct blind testing and statistical analyses. Behavioural data and fibres volume were analysed using sets of between and within orthogonal contrasts controlling the per contrast error rate at alpha = 0.05 (*Hays, 1963*). Simple effects analyses were conducted to establish the source of significant interactions. Statistical analyses were performed using PSY Statistical Program (*Bird et al., 2022*) and graphs were created using GraphPad Prism.

All experiments employed a between- × within-subjects behavioural design. In the first experiment, the between-subject factors were group (Pre vs. Post) and treatment (CTL vs. SAP) and the within-subject factors were training day (acquisition data) or devaluation for the test data (responding on lever associated with non-devalued or devalued outcome). In the 6-OHDA experiments, the between-subject factor was group (CTL vs. 6-OHDA or CTL vs. 6-OHDA+Desi) and the within-subject factor was training day (acquisition data) or devaluation (test data).

To analyse DβH and TH fibres volume, the between-subject factor was group (experiment 1: CTL vs. SAP; experiment 2: CTL, 6-OHDA+Desi, or 6-OHDA) and the within-subject factor was region (VO vs. LO) for the OFC. There was no within-subject factor for the quantification of fibres in the other regions of PFC. In the final chemogenetics experiment, the between-subject factor was treatment during reversal acquisition (vehicle vs. DCZ) and the within-subject factors were training day (acqui-sition data) or treatment during reversal test (vehicle vs. DCZ) and devaluation (test data). To analyse hM4Di/c-Fos colocalization, a Student's t-test was used to compare vehicle- and DCZ-treated animals within the stress condition.

## Acknowledgements

This work was supported by the French National Research Agency (CE37-0019 NORAD to EC and EJK) and by the Fondation pour la Recherche Médicale (FRM grant number ECO20160736024 to J-CC). Funding sources had no further role in study design, in the collection, analysis, and interpretation of data, in the writing of the report and in the decision to submit the paper for publication. Microscopy was completed at the Bordeaux Imaging Centre, a service unit of CNRS-INSERM and Bordeaux University and member of the national infrastructure, France BioImaging. The authors thank Hadrien Plat and Angélique Faugère for help with immunofluorescence and Yoan Salafranque (CNRS, INCIA, CNRS, UMR 5287) for expert animal care. The authors are grateful to Alain Marchand for his help in this project.

## Additional information

### Competing interests

Mathieu Wolff: Reviewing editor, *eLife*. The other authors declare that no competing interests exist.

### Funding

| Funder | Grant reference number | Author |
| --- | --- | --- |
| Agence Nationale de la Recherche | CE37-0019 NORAD | Eric J Kremer<br>Etienne Coutureau |
| Fondation pour la Recherche Médicale | ECO20160736024 | Juan Carlos Cerpa |

The funders had no role in study design, data collection and interpretation, or the decision to submit the work for publication.

### Author contributions

Juan Carlos Cerpa, Data curation, Funding acquisition, Investigation, Methodology, Writing – original draft; Alessandro Piccin, Data curation, Validation, Investigation, Methodology, Writing – review and editing; Margot Dehove, Investigation; Marina Lavigne, Validation, Methodology; Eric J Kremer, Conceptualization, Funding acquisition, Methodology; Mathieu Wolff, Conceptualization, Supervision, Writing – review and editing; Shauna L Parkes, Conceptualization, Formal analysis, Supervision, Writing – original draft, Writing – review and editing; Etienne Coutureau, Conceptualization, Supervision, Funding acquisition, Methodology, Writing – original draft, Project administration, Writing – review and editing

### Author ORCIDs

Alessandro Piccin ⓘ http://orcid.org/0000-0001-9566-3808
Mathieu Wolff ⓘ http://orcid.org/0000-0003-3037-3038
Shauna L Parkes ⓘ http://orcid.org/0000-0001-7725-8083
Etienne Coutureau ⓘ http://orcid.org/0000-0001-6695-020X

### Ethics

Experiments were performed in accordance with current French (Council directive 2013-118, February 1, 2013) and European (directive 2010-63, September 22, 2010, European Community) laws and policies regarding animal experimentation. The experiments received approval from the local Bordeaux Ethics Committee (CE50).

### Decision letter and Author response

Decision letter https://doi.org/10.7554/eLife.81623.sa1
Author response https://doi.org/10.7554/eLife.81623.sa2

## Additional files

**Supplementary files**
• MDAR checklist

**Data availability**
All data generated or analysed during this study are included in the supporting file.

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
