## [Editor Report]

The capacity to flexibly modify our actions in order to seek goals relies upon specific brain regions and neurochemicals. Here, Cerpa et al., identify norepinephrine (but not dopamine) within the ventrolateral orbitofrontal cortex (OFC) as key to updating identity-specific action-outcome associations when environmental conditions change. These conclusions are well supported by the data and will be of interest to behavioural neuroscientists studying the function of OFC or noradrenaline signalling, as well as researchers studying associative learning more broadly.

---

## [Decision Letter]

**Decision letter after peer review:**

Thank you for submitting your article "Noradrenergic signaling in the rodent orbitofrontal cortex is specifically required to update goal-directed actions" for consideration by *eLife*. Your article has been reviewed by 3 peer reviewers, including Laura A Bradfield as Reviewing Editor and Reviewer #1, and the evaluation has been overseen by Kate Wassum as the Senior Editor.

Essential revisions:

1) Additional reporting regarding the anatomical specificity of placements within orbitofrontal cortex (OFC) is necessary. Even small anatomical shifts can make big differences with regards to behavioural outcomes in this part of the brain, so it is important to be as clear as possible about the precise location of these placements across all subjects.

2) Reviewer #2 is concerned that the disruption of behavioural updating is not specific to noradrenaline signalling within the OFC and makes several suggestions of additional experimental work to address this concern. Please either complete the additional experiments, or quantify the anatomical effects of toxin experiments in neighbouring brain regions in addition to mPFC or, as a final option (although least preferred), claims of specificity of the effect to OFC could be toned down.

3) Some comment on the limitation of the exclusive use of males would be beneficial.

Please let me know if you will need additional time to complete these essential revisions. Please also respond to all other reviewer points.

*Reviewer #2 (Recommendations for the authors):*

In the absence of additional experiments, I would be comfortable with either a quantification of the anatomical effect of the toxin experiments in other regions outside OFC (not just mPFC) so long as only mPFC has a non-specific ablation, or a toning down of the claim of specificity of the effect in OFC.

Also, just to make sure, the IgG saporin that was used here was truly an inactive one and not the one targeting cholinergic fibers, right? It might be good to be explicit about this.

*Reviewer #3 (Recommendations for the authors):*

Specific comments: The only major experimental issue I have is that it would be nice to validate the DREADDs manipulations here, either with immunostaining for immediate-early gene expression or electrophysiological measures.

It would also not hurt to be more inclusive in the VLO behavioral studies being cited. Teams in addition to Balleine's and the present group have contributed to knowledge in this field.

Figures: Because the subregion of OFC being manipulated is pretty important in this line of research, it's critical to see zoomed-out infusion site images of the OFC throughout, plus representations of region of the OFC being sampled in microscopy efforts (for instance in figure 2). A crucial argument is that the VLO performs X,Y function, but almost none of the figures actually show images that allow us to confirm that we are in VLO and not the medial or lateral OFC regions. Relatedly, I'm not entirely clear what I'm looking at in Suppl. Figure 4B, left. The triangular shape in the center of the section looks like the very anterior portion of the corpus collosum, but that would be problematic because the OFC is no longer present at that anterior position. Further anatomical context would be helpful.

[Editors’ note: further revisions were suggested prior to acceptance, as described below.]

Thank you for resubmitting your work entitled "Noradrenergic signaling in the rodent orbitofrontal cortex is specifically required to update goal-directed actions" for further consideration by *eLife*. Your revised article has been evaluated by Kate Wassum (Senior Editor) and a Reviewing Editor.

The manuscript has been improved and many of our concerns addressed, but there are two areas in which the reviewers and myself felt more could have been done to allay our concerns. The remaining issues are outlined below:

1. In our original decision letter, all three reviewers requested more information regarding the anatomical specificity of the circuit manipulations and therefore the role of this noradrenergic->OFC pathway in updating goal-directed actions. Although you have added some more diagrams and images to this effect, we had been expecting some representation extent of lesion/expression across subjects and possibly some additional microscopy representations etc which were not provided. Moreover, we asked for additional work to quantify the effect of toxins in regions adjacent to OFC, which the authors have done, however, according to these figures, saporin and 6-OHDA (respectively) DO appear to have had off-target effects. As in, they appear to have reduced the volume of fibres in medial OFC, and M2, and for saporin, also in the insula. The authors have added a passage in the discussion noting that the anatomical specificity is not ideal and that collaterals in other regions could contribute to the effects, which is sufficient here, but these results do call into question the strength of the overall conclusions of the specificity of this behaviour to this particular pathway. Therefore, we think it is important to include a representation of extent of lesion/expression across subjects for each experiment, e.g., with a map representing this for each subject overlaid, or at the least minimal/maximal spread. We think this will be critical for readers to understand the extent of anatomical specificity. We also request some toning down of claims in the title and abstract, e.g. "Together, our results indicate that noradrenergic…" could become "Together, our results suggest that noradrenergic…".

2. The authors decided not to offer any immunohistochemical, electrophysiological, or another type of validation of their chemogenetic manipulations as requested. Given that the specific manipulation used here is novel, some validation of these manipulations (other than their effects on behaviour) would have been ideal. We thought we would give the authors another opportunity to provide this, but if the authors strongly feel that it is unnecessary then please state this explicitly in the discussion (i.e. "we did not feel that some independent validation of our chemogenetic manipulations was necessary because…") and provide adequate reference to prior validation of this approach.

---

## [Author Response]

Essential revisions:1) Additional reporting regarding the anatomical specificity of placements within orbitofrontal cortex (OFC) is necessary. Even small anatomical shifts can make big differences with regards to behavioural outcomes in this part of the brain, so it is important to be as clear as possible about the precise location of these placements across all subjects.

Precise anatomical locations are now provided in Figure 1B and Figure 2A, as well as Figure 1—figure supplement 1A and Figure 2—figure supplement 1A.

2) Reviewer #2 is concerned that the disruption of behavioural updating is not specific to noradrenaline signalling within the OFC and makes several suggestions of additional experimental work to address this concern. Please either complete the additional experiments, or quantify the anatomical effects of toxin experiments in neighbouring brain regions in addition to mPFC or, as a final option (although least preferred), claims of specificity of the effect to OFC could be toned down.

As requested, we now provide additional quantification of noradrenergic fiber loss in neighbouring regions known to be involved in goal directed behavior, namely the insular cortex (e.g., Balleine and Dickinson, 2000; Parkes and Balleine, 2013), the medial orbitofrontal cortex (e.g., Bradfield et al., 2015; Gourley et al., 2016) and secondary motor cortex (Gremel et al., 2016). These quantifications are shown in two additional supplemental figures (Figure 1—figure supplement 1B and Figure 2—figure supplement 1B).

3) Some comment on the limitation of the exclusive use of males would be beneficial.

The choice to use male rats only was primarily dictated by the known anatomical and physiological variations of the LC between male and female animals (Joshi and Chandler, 2020). We have now stated in the abstract that only males were used and have also added the following statement to the manuscript:

Page 19, line 590: “We must also acknowledge that only male rats were used in the current study. The LC displays some anatomical and physiological variations between male and female rats (Joshi and Chandler, 2020), therefore a thorough characterization would also need to integrate this sex factor.”

Reviewer #2 (Recommendations for the authors):In the absence of additional experiments, I would be comfortable with either a quantification of the anatomical effect of the toxin experiments in other regions outside OFC (not just mPFC) so long as only mPFC has a non-specific ablation, or a toning down of the claim of specificity of the effect in OFC.Also, just to make sure, the IgG saporin that was used here was truly an inactive one and not the one targeting cholinergic fibers, right? It might be good to be explicit about this.

Yes, the IgG saporin used was Mouse IgG-SAP (Cat. #IT-18, Advanced Targeting Systems). This has been clarified in the results and the Key Resources Table.

Reviewer #3 (Recommendations for the authors):Specific comments: The only major experimental issue I have is that it would be nice to validate the DREADDs manipulations here, either with immunostaining for immediate-early gene expression or electrophysiological measures.

While we do not provide validation of the DREADDs manipulation using immediate early gene expression or electrophysiology, we believe that there is nonetheless sufficient evidence for the efficiency of these tools. First, our immunohistochemical analysis clearly demonstrates that the CAV2 vector with the PRS promotor specifically targets NA neurons in the LC and that these neurons express the inhibitory DREADDs. Second, previous studies have validated the use of the CAV2-PRS construct using a range of actuators, including excitatory and inhibitory opsins, as well as potassium channels (Howorth et al., 2009b, 2009a; Hickey et al., 2014; Li et al., 2016). Third, the behavioural dissociation between outcome identity reversal (this study) and Pavlovian contingency degradation (ongoing study, see Figure 1) suggests a subtle prefrontal coordination of NA dependent processes. Finally, DCZ as a ligand has now been proved to be highly efficient in a range of species, including rodents (Nagai et al., 2020; Nentwig et al., 2021; Oyama et al., 2022).

It would also not hurt to be more inclusive in the VLO behavioral studies being cited. Teams in addition to Balleine's and the present group have contributed to knowledge in this field.

We agree and apologize for this oversight. Additional relevant studies are now cited, including Zimmerman et al., 2017; 2018; Whyte et al., 2019; Gremel and Costa, 2013; Gremel et al., 2016; Schreiner and Gremel, 2018; Rhodes and Murray, 2013; Fiuzat et al., 2017; Malvaez et al., 2019; Sias et al., 2021; Wassum, 2022.

Figures: Because the subregion of OFC being manipulated is pretty important in this line of research, it's critical to see zoomed-out infusion site images of the OFC throughout, plus representations of region of the OFC being sampled in microscopy efforts (for instance in figure 2). A crucial argument is that the VLO performs X,Y function, but almost none of the figures actually show images that allow us to confirm that we are in VLO and not the medial or lateral OFC regions.

The regions being sampled in microscopy images and quantification of noradrenergic and dopaminergic fibres are now shown in in Figure 1B and Figure 2A as well as Figure 1—figure supplement 1A and Figure 2—figure supplement 1A.

Relatedly, I'm not entirely clear what I'm looking at in Suppl. Figure 4B, left. The triangular shape in the center of the section looks like the very anterior portion of the corpus collosum, but that would be problematic because the OFC is no longer present at that anterior position. Further anatomical context would be helpful.

We have now updated this figure to provide clearer anatomical context (Figure 4—figure supplement 1).

[Editors’ note: further revisions were suggested prior to acceptance, as described below.]

Reporting of the anatomical specificity of the behavioural effects.

We have included new figures with a more specific and exhaustive description of the anatomical boundaries of the manipulations. For Experiments 1 and 2 we now show photomicrographs of maximum and minimum NA fiber loss in vlOFC and A32 (see Figure 1—figure supplement 1 and Figure 2—figure supplement 1) and also maps illustrating fiber loss (see Figure 1 and Figure 2). For Experiment 3, we now provide a map illustrating viral expression across subjects in vlOFC and A32, as well as representative photomicrographs of the targeted areas (see Figure 4B).

2. Toning down of the claims

As requested, we have toned down our claims about the specificity of our interventions, including changing the title and modifying phrases in the abstract and discussion (highlighted in yellow).

3. Validation of the viral construct

We now provide validation of the viral construct and DREADD ligand, as described in the Methods section (from line 755). Briefly, we quantified the number of hM4Di-expressing cells that were c-Fos positive following injection of the DREADD ligand DCZ *versus* injection of vehicle following a stress procedure (foot-shocks). We found that rats given DCZ had significantly fewer hM4Di/c-Fos-positive cells in the locus coeruleus (≈ 35% decrease) compared to vehicle-injected rats (see Figure 4—figure supplement 1).